

# Optimization analysis of active solar still using design of experiment method

Mohammad Omar Abu Abbas, Malik Yousef Al-Abed Allah[*,] Qais Nedal Al-Oweiti
Department of Mechanical Engineering, Jordan University of Science and
Technology, Irbid, Jordan
*Correspondence: myalabedallah16@eng.just.edu.jo
**Mohammad Omar Abu Abbas,** Department of Mechanical Engineering, Jordan
University of Science and Technology, 22110, Irbid, Jordan,
moabuabbas16@eng.just.edu.jo
**Malik Yousef Al-Abed Allah,** Department of Mechanical Engineering, Jordan
University of Science and Technology, 22110, Irbid, Jordan,
myalabedallah16@eng.just.edu.jo
**Qias Nedal Al-Oweiti**, Department of Mechanical Engineering, Jordan University of
Science and Technology, 22110, Irbid, Jordan, qnaloweiti16@eng.just.edu.jo





**Key words:** solar still, DOE, factorial design, thickness, productivity,
water depth, insulation.

## Abstract:

Mathematical model for different configurations of active solar still has been
analyzed. Theoretical analysis of energy balance for the active solar still components
has been developed. A statistical manner for examination, evaluation, and optimizing
the performance of the active solar distillation system with known input factors has
been performed using the Design of Experiments (DOE) method. Some processes
with input variables (factors) and predicted output variables (responses) have been
evaluated. Input factors influencing the responses have been identified. The impact of
each variable (factor) and integration of two factors at the same time (called
interactions) have been estimated. Influences of various factors on a particular study
at a time rather than performing different separated studies have been investigated. 11
variables (basin area, depth of saline water, external power, air blowing system,
condenser material, condenser thickness, condenser area, insulation thickness,
insulation material, ambient air temperature, and make-up water system ) have been
studied to show their effects on three responses (mass output, saline water temperature
and condenser cover temperature). The statistical results showed that the most
significant factors affected on mass output (distilled water), respectively, were the
external power, the depth of the saline water and the basin area of the active still.
While the most influence factors affecting the saline water temperature and the
condenser cover temperature were the depth of saline water, external power and air
blowing system respectively.









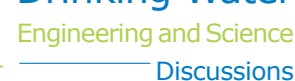

**Nomenclatures**:
A        Area (m$^2$)
$cp$      Specific heat of material (J/kg.k)
$m$       Amount of saline water (kg)
P        External power (W/m$^2$)
$Q_{cb-w}$    Convection heat transfer from basin plate to saline water (W)
$Q_{cb-w}$    Convection heat transfer from saline water to condenser (W)
$Q_{rw-c1}$    Radiation heat transfer from saline water inner condenser (W)
$Q_{ew-c1}$    Evaporation heat transfer from saline water to inner condenser (W)
$Q_{cc2-a}$    Convection heat transfer from outer condenser cover to ambient (W)
$Q_{rc2-s}$    Convection heat transfer from outer condenser cover to ski (W)
$Q_{cnc1-c2}$      Conduction heat transfer from inner condenser cover to outer condenser
(W)
$Q_{loss-ba}$    Conduction heat transfer from basin plate to ambient (W)
$Q_{mw}$      Make-up saline water (W)











## 1. **Introduction**:

Water is an essential component of human health. Nearly 60% of the human body is composed of water. It is important to note that the individual's need for water varies from person another depending on the nature of the individual's daily physical activities and the drought proportion in the place where they live. Therefore, the individuals tend to drink sufficient amounts of water to prevent them from the drought. Consequently, it leads to drain the body's energy, and cause tired. The National Academy of Sciences has determined the amount of water that is recommended daily, namely 3.7 liters of water for males and 2.7 liters of water for females. In fact, these amounts include water obtained from drinking water, and eating other foods and beverages. Although three-quarters of earth is covered with water but, the clean water does not exceed 2.75%, which is a low proportion comparing with saltwater.

Improving the performance of solar still depends mainly on decreasing condenser cover temperature and increasing saline water temperature. Enhancing the productivity of solar still has received significant attention from many researchers. The daily production of solar still depends on several factors such as climatic conditions (solar radiation intensity, ambient temperature, and wind speed), condensation surface inclination, insulation type and thickness, solar still geometry, the orientation of still and depth of salty water.

Bataineh and Abu Abbas (2020). studied numerically the effect of solar still productivity by adding vertical fins, external reflectors and both of them together at different seasons. The theoretical results show that the productivity has not been affected significantly by adding fins and the efficiency of still increase by 13%, 20%, 28%, 33%, 37% and 46% in June, April, September, October, January, and December respectively when adding external reflectors. Bataineh and Abu Abbas (2020). investigated theoretically and experimentally, the effect of single sloped solar still performance when adding $Al_2O_3$ and $SiO_2$ nanoparticles. The results show that the productivity of still boosted by 10% and 8.5%, respectively, at 0.005 m saline water depth and 0.2% concentration of nanoparticles. Manokar et al (2020). Analyzed the performance of pyramid solar still at different saline water thickness, solar still with insulation material and solar still without insulation material. The experimental results inferred that the performance of still increase as saline water depth decrease and the productivity of still is improved 113 by integrate insulation material in the still. Khalifa et al (2009). Verified the effect of insulation thickness (3, 6 and 10 cm) on the efficiency of solar still. The experimental results described that the productivity of still increase as insulation thickness increase up to specific value (6 cm) beyond which the effect of increasing thickness become insignificant. Abu Abbas and Al-Abed Allah (2020) examined numerically the impact of condenser materials type and condenser incline on the performance of the solar still. The results reveal that the daily

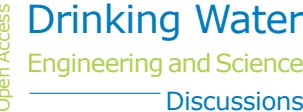

solar still productivity increases as transmissivity value of condenser material increase. In addition, it was noted that the maximum productivity in summer (May) was at the lowest condenser slope angle (5°) and it was decreased as the condenser slope angle increased. On the other hand, the maximum productivity of solar still in the winter season (January) was at (20°) and then decreased as the condenser slope angle increased. Dubey and Mishra (2019) examined the influence of three glass cover angles (15˚, 30˚, and 45˚) on solar still productivity. They found that the maximum productivity was obtained at 15˚ tilt angle which was nearer to the latitude of Raghogarh, Guna. Kumar et al. (2008) examined the V-type solar still with floating charcoal absorber over the saline water in M.S.basin and with and without the boosting mirror. The yield increases with boosting the mirror, but overall efficiency reduces due to an increase in loss and condensate could be easily collected because of the collection at the center. Madhlopa et al. (2009) found out that utilizing multi evaporators and multi condensers have improved the solar still performance by 62%. Hansen et al. (2017) enhanced solar still productivity by using fin shaped absorber configuration. Their results showed that the solar still efficiency increased by 25.75%. E. Kabeel et al. (2018) investigated the effect of utilizing a different type of phase change materials (PCM) to enhance solar still performance. The theoretical results showed that the A48 type of PCM has the highest increase in efficiency reach up to 92%. Al-harahsheh et al. (2018) conducted an experimental study on single slope solar still integrated with phase change material (PCM) and connected with a solar water collector to enhance basin water temperature of solar still. Zurigat et al. (2004) studied the effect of a regenerative concept on solar still performance. Their results illustrated that the performance of regenerative still concept is higher by 20% compared with conventional solar still. Nisrin Abdelal et al. (2018) conducted an experiment to study the effect of using absorber plates made of carbon fiber/nanomaterials-modified epoxy composites at different concentrations. Their results show that the productivity of still increase by 109% and 65% when adding 5% and 2.5% Nano weight concentrations respectively. Agrawal et al. (2017) conducted experimental and theoretical study to investigate the effect of saline water depth (2 cm, 4 cm, 6 cm, 8 cm and 10 cm) on solar distillation system productivity. Their results illustrated that the distilled water of solar distillation system increases as decreasing water depth. Hitesh et al. (2012) examined the effect of floating plates (such as galvanized iron and aluminum) on solar still productivity. It was observed that the aluminum plate enhanced the productivity of still more than galvanized iron plate.

Design of Experiment is an efficient tool for increasing the quantity of data gained from a study in addition to reducing the amount of data to be obtained, which, in this case is decreasing the number of trial runs. It should be remarked that all of the researches have studied the influence of utilizing one parameter at a time while keeping the other parameters fixed will not occur to understand the interaction. Here

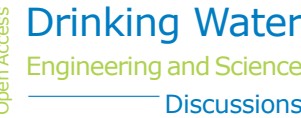

in this research, we collected all the parameters that could affect the active solar still
system to show which parameters have the most significant effect and which of them
does not has any influence when they are being together at the same time. Moreover,
to explain the interaction between the most significant factors and their regression
equations. In addition to highlight on the most important factors that create the
optimal design for active solar still system.

**2. Methodology**:

**2.1 Description:**
The main components of active solar distillation system are shown in Fig. 1. The
water tank is used as a make-up water system to compensate purified water. An
external power device is used to heat the basin plate. Large proportion of heat will
transfer by convection to the saline water while the rest of it will be lost outside by
conduction through the bottom and the sides of still. The heat will be conveyed from
the high saline water temperature to the internal surface of cooled condensation cover
by evaporation, convection and radiation. The heated saline water will convey heat to
the inner cooled condensation cover by evaporation, convection and radiation. Then
part of heat will be transferred by conduction between two sides (from the inner to the
outer surface) of the condenser, and by radiation and convection from the upper
surface of the condenser to the surrounding air. Inclined condensation cover is used to
move evaporated water to the water collector. Bottom and all sides of solar distillation
system have a specific insulation material with a proper thickness to eliminate heat
losses from heated saline water to the surrounding. Moreover, Fig. 2a and Fig. 2b
show solar still with increasing condensation cover area and adding fan respectively
to enhance convection heat transfer from upper surface of inclined surface to the
ambient air. as a result, increasing condensation rate. Fig. 3 shows distilled water
cycle for solar distillation system.







## 2.2 Mathematical model:



A complete non liner differential equations model that shows the heat transfer and
energy processes in the main components of the active solar distillation system has
been written. These equations helped to calculate the quantity of the distilled water
temperature and the condenser cover temperature at any time and at different system
configurations. The theoretical results were founded by solving the main energy
balance equations for the basin plate, saline water, the inner and the outer condenser
covers of the active solar distillation system. The saline water, the basin plate, the
inner and the outer condenser cover temperatures were evaluated every 5 hours to
show the effect of changing different parameters on the solar distillation system
productivity. The numerical model was solved by Matlab software. Energy balance
equations for main solar still components are presented as follow:

As shown in Eq. (1), fraction of the external power connected with the solar
distillation system is transmitted to the basin plate as heat and then it is transferred to
saline water by convection. Other amount of energy is lost to the ambient through
bottom insulation material by conduction.


$$P_t A_b = m_b cp_b \frac{dT_b}{dt} + Q_{cb-w} + Q_{cb-ba} \tag{1}$$


The transient energy balance equation for the saline water is given as Eq. (2),
fraction of heat is transmitted to saline water by convection. All heat gained is lost in
two approaches; specific quantity of energy is stored in saline water due to its specific
heat property. The rest of energy is released to the inner condenser cover by
evaporation, convection and radiation.


$$Q_{cb-w} = m_w cp_w \frac{dT_w}{dt} + Q_{cw-c1} + Q_{ew-c1} + Q_{rw-c1} + Q_{mw} \tag{2}$$


Energy balance equation for the inner condenser cover is presented as Eq. (3). The
heat energy arrived from saline water surface is absorbed by the inner condenser
cover and then released by conduction through thickness of the cover.

$$Q_{cw-c1} + Q_{ew-c1} + Q_{rw-c1} = m_c cp_c \frac{dT_{c1}}{dt} + Q_{cc1-c2} \tag{3}$$


Energy balance equation for the outer condenser cover is shown as Eq. (4). The
heat lost by conduction to the outer condenser cover is transferred by convection to
the air and by radiation to the sky.

$$Q_{cnc1-c2} = m_c cp_c \frac{dT_{c2}}{dt} + Q_{rc2-sk} + Q_{cc2-a} \tag{4}$$



### 2.3 Design of Experimental:

Design of Experimental is a valuable tool for researchers and designers which used to develop any system design. This tool can reduce designing time and cost with high reliability than other designing approaches. As it is known, the main purpose of conducting an experiment is to be found which system parameters have most significant on the specific response (output of the system). Using this tool, it will be known the effected factors that improve the system and neglect the most fewer effected factors.

In this study, factorial design has been used to determine the most influence and not influence of 11 factors, interaction between them and regression equations for designing solar distillation system. Three responses have been evaluated which are distilled water, saline water temperature and inner condenser temperature.

### 2.3.1 Factorial Design:

A factorial design is an important type of design of experiments approaches. It is used to determine the most effected parameters to find the optimal design for the system of interest. Therefore, a huge time and tremendous effort could be saved instead of applying a full-scale simulation. Furthermore, the most valuable advantage of the factorial design is to find the regression equations and interactions between the factors that would be impossible to calculate in the other analysis approach. In order to achieve all the previous advantages, the factorial design method could set two values for each factor (levels), these levels and their values is determined by experience, then the researcher has to create configuration runs table using Minitab software according to probability counting rule ($2^k$) where k is the number of factors each one has two levels (+1 value for a high level and –1 value for a low level.). Table. 1 below displays the main factors of interest

### 2.3.2 Reduced Factorial $2^{(11-4)}$

The main purpose of reduced factorial design that the system is performed with much less trials by sacrificing interactions for more than three factors. The reduced factorial which has been selected is $2^{(k-r)}$ where r refers to number of reduced factors. Moreover, reduced factors have been chosen very carefully by checking the alias structure, resolution, balancing and orthogonally. In this study a $2^{(11-4)}$ reduced factorial has been used with V resolution, which means that the main effects and two-way interactions not confounded except with higher order interactions. Matlab has been used to simulate the suitable and necessary simulations and Minitab to investigate the main influence factors and interactions between them with high accuracy.





## 3. Numerical simulation assessment

Fig. 4 shows the flowchart used to evaluate the most significant factors that have impacts on solar distillation system. The simulation starts with Minitab program to find the number of the solar still configurations using 11 factors. Determine type of analysis (reduced or full factorial), factors number and nature of runs (randomize or non-randomize runs) are the important steps in this software. Furthermore, a numerical model was written using Matlap program to analyze the effect of the solar still configurations calculated using Minitab program. Minitab is computer software which was developed to solve a mathematical model of the still components (condensation surface, saline water and basin plate) for different solar still configurations. The Temperature of the condensation cover, saline water and the basin plate were founded by solving the numerical model using Runge–Kutta method. All still components' temperatures and purified water were founded every 5 hours. Initial temperature values of different components of the solar still were equaled the ambient temperature value. Using these initial temperatures, the condensation cover, saline water and the quantity of distilled water were calculated. The procedures were repeated for every solar still configuration (run) which was taken from Minitab program. Finally, all solar still configurations results that calculated from MATLAP were analyzed using Minitab program to show their effects.

## 4. Results:

The results of mathematical and designing calculations could discover effect of different factors on active solar still responses. Three responses have been studied: amount of distilled water (mass output), water temperature, and condenser cover temperature. External power, basin area, water depth, insulation material, insulation thickness, condenser material, condenser area, thickness of condenser, air blowing system, Make-up water system, and ambient temperature are considered as variables to understand their influences on the mentioned responses. To be more effective, the simulation results were gained based on the design of experiment approach (DOE). The (DOE) was conducted using a reduced factorial method to show their direct effects, their interactions, and the optimization design for the system.

### 4.1 Main effect plots on the responses:

Fig. 5a, Fig. 5b and Fig. 5c showed the main factors influenced on the responses of active solar still system. The x axis shows responses values while the y axis shows


the high and the low levels of the factors. It was clearly noted that, as inclination of
the lines increase, the effect of the factors on the responses will be significant. The
results found that the most important factors that enhance mass output are amount of
external power, water depth, and basin area respectively. Where the mean mass output
recorded at the high and low levels were 3.02 L and 1.24 L respectively for external
power factor and 1.3L and 2.8L respectively for water depth factor. While, it is
reached to about 2.8L and 1.4L at high and low levels of the basin area respectively.
Moreover, other factors have a little effect on the system. Furthermore, the simulation
results indicated that the water depth, the amount of external power, the air blowing
system, and the condenser material respectively are the main factors that have the
most influence on the water temperature and condenser cover temperature of the
system while rest factors have a little effect on it as shown in Fig. 5b. and Fig. 5c.


**4.2 Interaction effect plots:**
The independent variables (factors) might interact with each other. It happens
when the influence of one factor depends on the value of another factor. Moreover,
the Interaction effects show that a third variable affects the relationship between an
independent and dependent factor (responses). This kind of scheme represents the fit
values of the dependent factor on the y-axis while the x-axis displays the values of the
first independent factor while the different lines describe the values of the second
independent factor. About the interaction schemes, parallel lines show that there is no
interaction between the two factors while the crossed lines and the lines that will be
crossed infer that there is an interaction effect between the factors. Here are the
figures for the factors that produced an interaction between each other for various
responses. Fig. 6a showed that the interaction effect on mass output. It was clearly
noted that (basin area*external power), (basin area*depth of water), (depth of
water*external power), (depth of water * air blowing system) and (condenser material
*depth of water) respectively have the greatest interaction effect between each other.
For example, the scheme for (basin area*external power) explains that mass output
level was higher when the external power and the basin area values were high.
Conversely, the maximum mass output have been achieved when the external power
and the basin area values were low. Fig. 6b showed effect of the interaction on water
temperature of the active solar still .it was shown that the highest interaction to
produce maximum water temperature were between (depth of water * air blowing
system), (condenser material *depth of water), (depth of water*condenser area),
(external power * air blowing system) and (depth of water*external power)
respectively. While the interaction plot affected on condenser temperature was
described in Fig. 6c. Whereas the important interaction effect were (depth of water *
air blowing system), (condenser material *depth of water), (power * air blowing

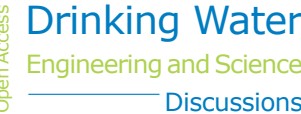

system), (depth of water*condenser area) and (depth of water*external power)
respectively.

## 4.3 Pareto charts of the standardized effects:


Fig. 7 display the Pareto charts of the standardized effects for various responses.
These charts determine the order of the most significant factors including main and
interaction factors that effect on the response's values. It is clearly observed that the
most influential factors on mass output are external power, depth of water, and basin
area respectively. While in the water temperature and condenser cover temperature,
the factors that have most significant effect are depth of water, external power, and air
blowing system respectively.

## 4.4 Regression equations:

Regression has been conducted on the results of factorial to show the effects of
these factors on the responses values. Eq. (5), Eq. (6), and Eq. (7) are the regression
functions predicted from the reduced factorial study which find that the highest and
lowest factors affected on three responses: distilled water, saline water temperature
and condenser cover temperature respectively. The constant numbers refer to the
factors affected ratio while the signals +, - refer to the high or low levels of the
factors.

$$
\begin{aligned}
\text{Mass} = {} & -1.026 - 0.0349\,A - 8.1\,B + 0.480\,C + 17.52\,D + 0.0809\,E + 4.67\,F \\
& - 0.0715\,G + 0.000990\,H - 0.1068\,J + 0.00196\,K - 0.1711\,L \\
& + 2.406\,A{*}D - 23.92\,C{*}D + 0.005022\,C{*}H - 0.02169\,D{*}H \\
& + 3.194\,D{*}J + 1.554\,D{*}L
\end{aligned}
$$

$$(5)$$


$$
\begin{aligned}
\text{Tw} = {} & 16.72 + 4.36\,A + 3386\,B + 17.19\,C - 10.7\,D - 3.52\,E + 41.5\,F \\
& - 0.627\,G + 0.04329\,H - 4.11\,J + 0.0179\,K - 0.761\,L \\
& - 759\,A{*}B + 1.166\,A{*}E - 0.00571\,A{*}H - 2617\,B{*}C - 13448\,B{*}D \\
& + 58.2\,D{*}E - 0.1492\,D{*}H + 80.9\,D{*}J - 0.00433\,E{*}H + 1.545\,E{*}J - \\
& 0.00675\,H{*}J
\end{aligned}
$$

$$(6)$$



$$
\begin{aligned}
\text{Tc} = {} & 10.21 + 3.61\,A + 2095\,B + 0.70\,C + 97.3\,D - 3.20\,E + 50.4\,F \\
& - 0.397\,G + 0.04501\,H - 3.61\,J + 0.0436\,K - 1.013\,L - 1203\,A{*}B \\
& + 77.4\,A{*}D - 0.01053\,A{*}H - 1.815\,A{*}J - 18424\,B{*}D + 60.7\,D{*}E - \\
& 0.2414\,D{*}H + 92.2\,D{*}J - 0.00717\,E{*}H + 1.633\,E{*}J - 0.01207\,H{*}J
\end{aligned}
$$

$$(7)$$







**4.5 Optimization Design:**


The designers should create the system by selecting the value of the optimal
factors that could enhance mass output. As mentioned above, the maximum water
output produced from the solar still could be achieved through increasing the saline
water temperature and decreasing the condenser cover temperature. Table. 2 and 3 list
the fit values and optimal design selected respectively, to achieve the optimal value
for the mass output, saline water temperature and condenser cover temperature.

**5. Conclusion:**


The results of theoretical and statistical analyses of 11 factors on the
active solar still system could be summarized as follows:
• The most important factors that can cause increasing in the mass output are the
amount of external power, water depth, and the basin area respectively.
• The thickness of the condenser and the ambient air temperature do not affect
the mean productivity
• Water depth, the amount of external power, the air blowing system, and the
condenser material, respectively, are the main factors that have the most
influence on the water temperature of the system.
• (Basin area*power), (basin area*depth of water), (depth of water*power),
(depth of water * air blowing system) and (condenser material *depth of
water), respectively, have the greatest interaction effect between each  other
that influence on mass output
• The significant interaction affected on saline water and the condenser
temperatures are (depth of water * air blowing system), (condenser material
*depth of water), (power * air blowing system), (depth of water*condenser
area) and (depth of water*power) respectively.
• The optimal design for the system can be attained ~~is~~ by selecting:
▪ Higher external power, basin area, condenser thickness, ambient
temperature and insulation thickness.
▪ Lower condenser area and depth of water.
▪ Using steel condenser material and fiberglass insulations rather than
any other materials.
▪ Adding air blowing system and removing make-up system.

**Conflict of Interest**


The authors declare that they have no conflict of interest.





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

471 1072.
















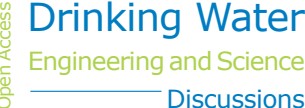















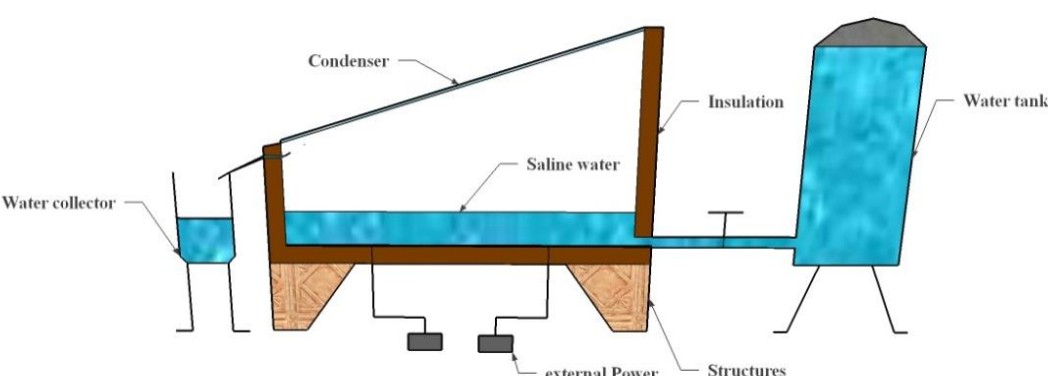

**Figure 1.** Solar distillation system.

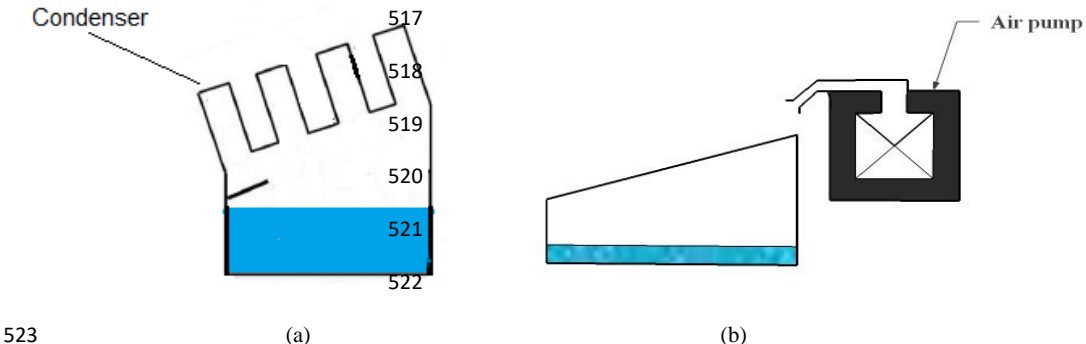

523                    (a)                                        (b)

**Figure 2. (a)** increasing condensation cover area and **(b)** adding fan to solar still

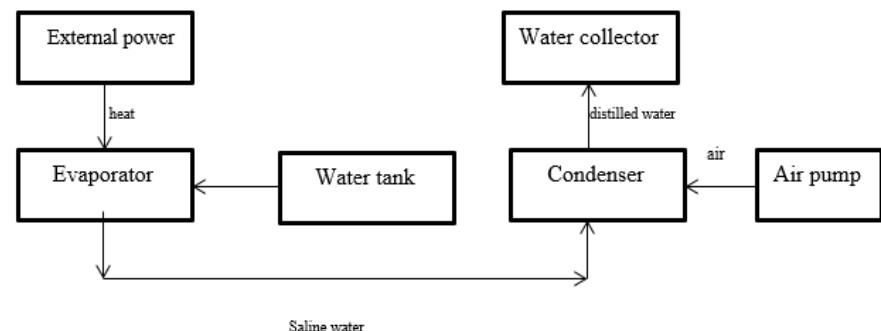

**Figure 3.** Distilled water cycle system.



**Table 1:** Description of factor levels.
**Table 2.** Responses fit values
**Table 3.** Values for optimal solar still design

Start calculation

Select analyse
type(reduced factorial or full factorial)

transfer all still configuration runs from
minitab to matlab

set system parameters (specification, dimensions and initial
condition) in matlab according to every minitab run

Set Time of simulation (h) and time step
size

No

Yes

Time =0
If time >= (h) thenTime=time+step size

End

Analyse factorial design using
minitab program

Calculate runge_kutta functions k1b, K2b, k3b, k4b using
Eq. 1

Calculate runge_kutta functions k1w, K2w, k3w, k4w using
Eq. 2

Calculate runge_kutta functions k1g, K2g, k3g, k4g using
Eq. 3

Calculate Tw,
Tg1,Tg2 and Tb

Calculate evaporation, conduction,
radiation heat transfer, water
productivity


**Figure 4.** System flow chart




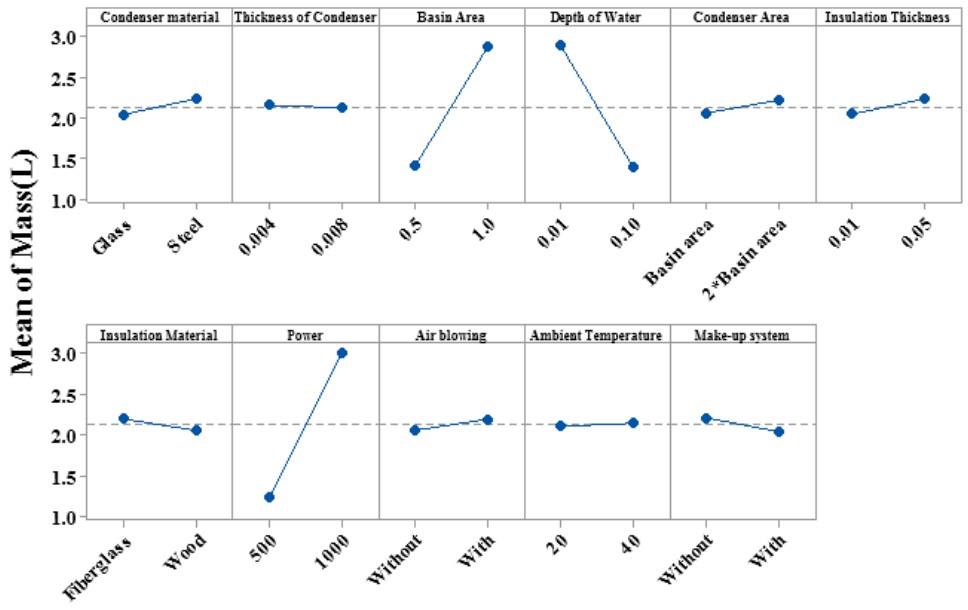


534                                                    (a)


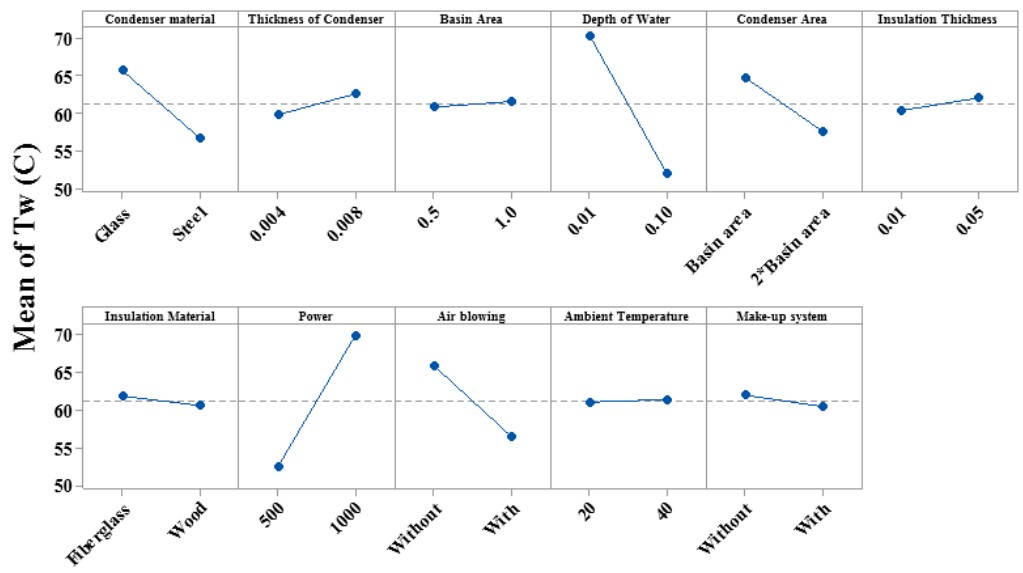



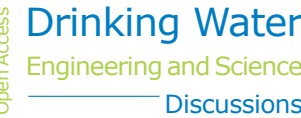

(b)

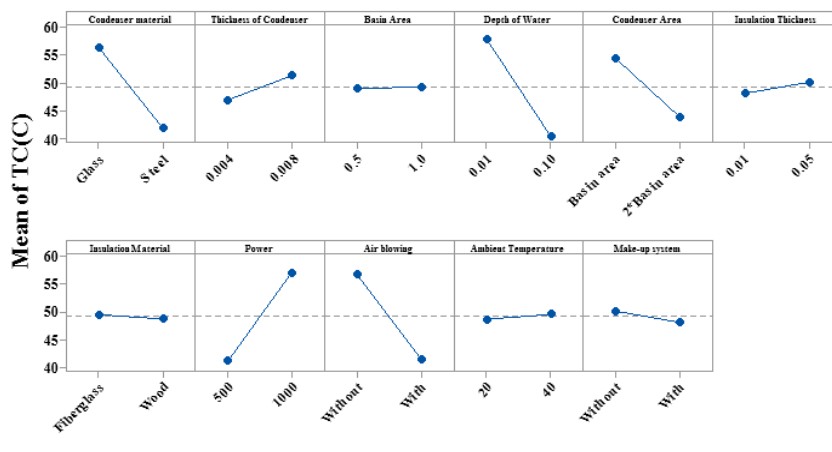


(C)

**Figure 5.** main effect factors on **(a)** mass output, **(b)** water temperature and **(c)** condenser cover temperature.



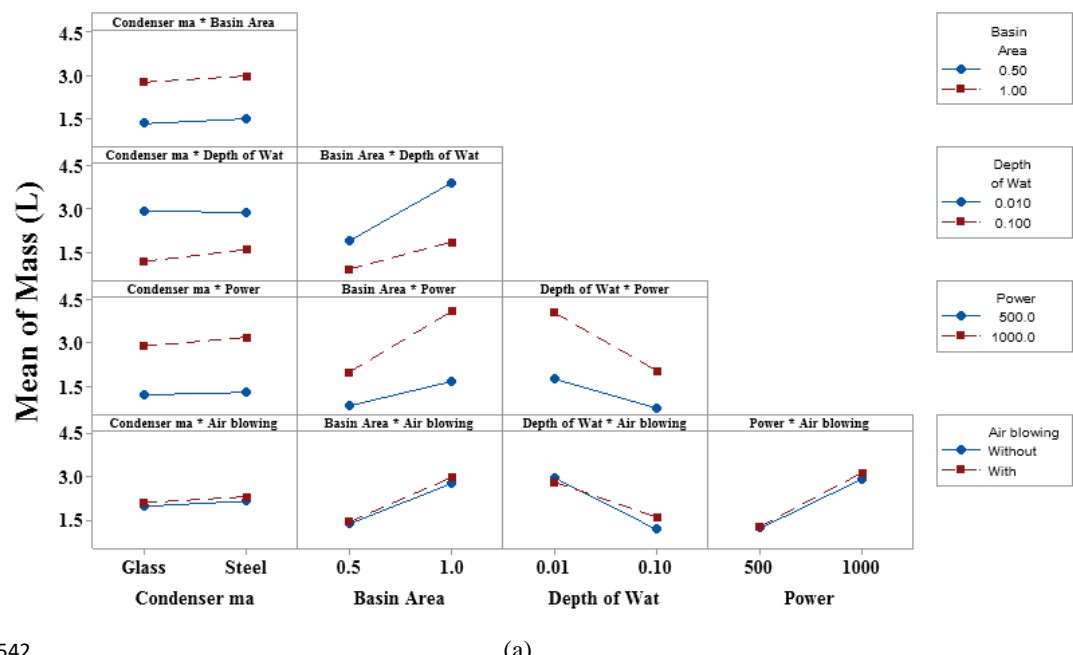

542                                                   (a)

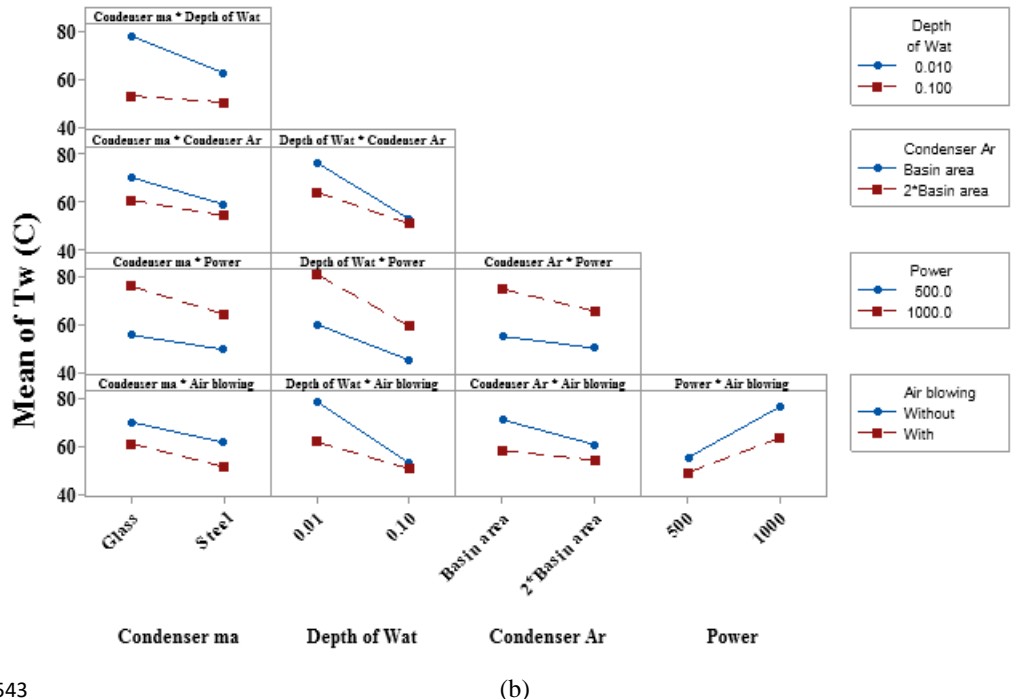

543                                                   (b)



544

545

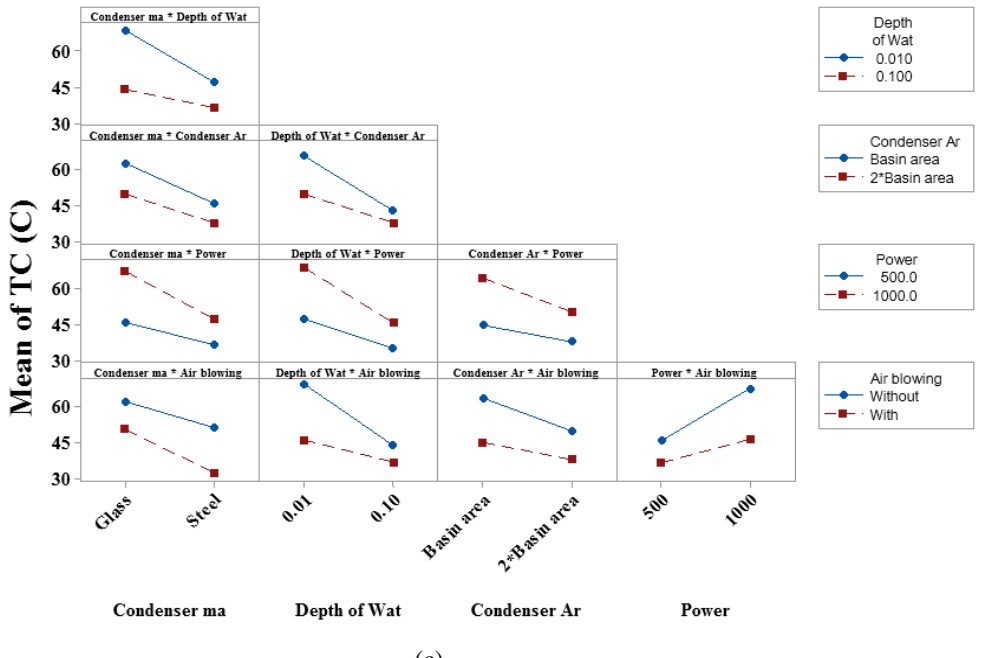

(c)

**Figure 6.** Interaction effect plot on **(a)** mass output, **(b)** water temperature and **(c)** condenser cover temperature.













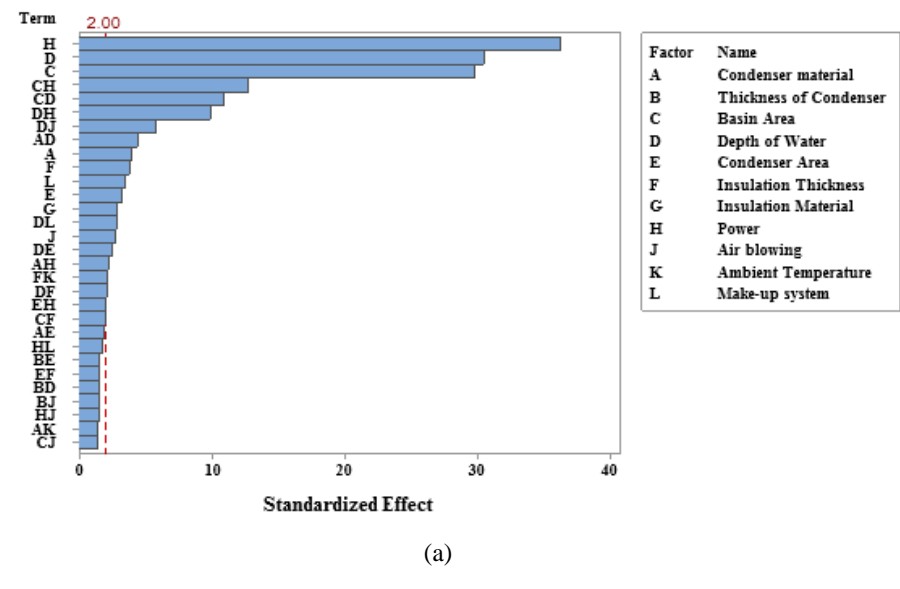

(a)


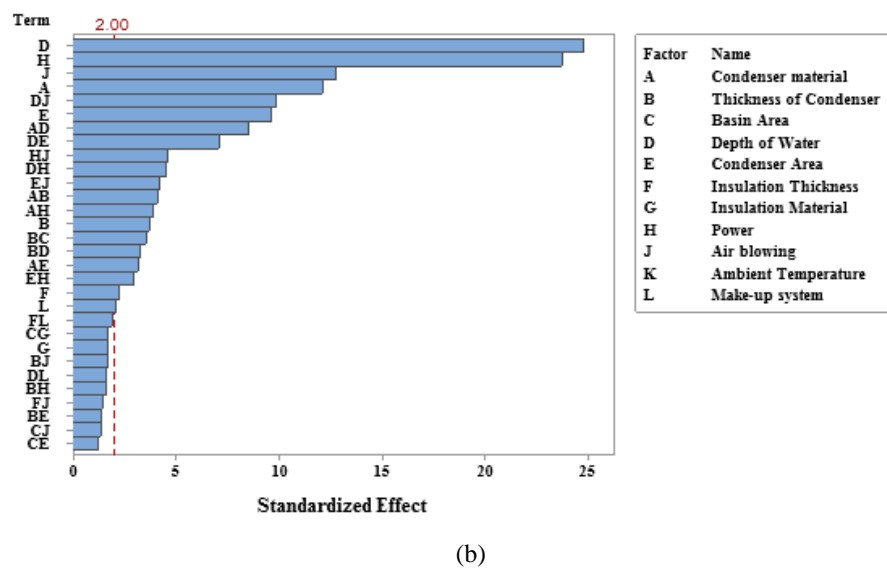

(b)

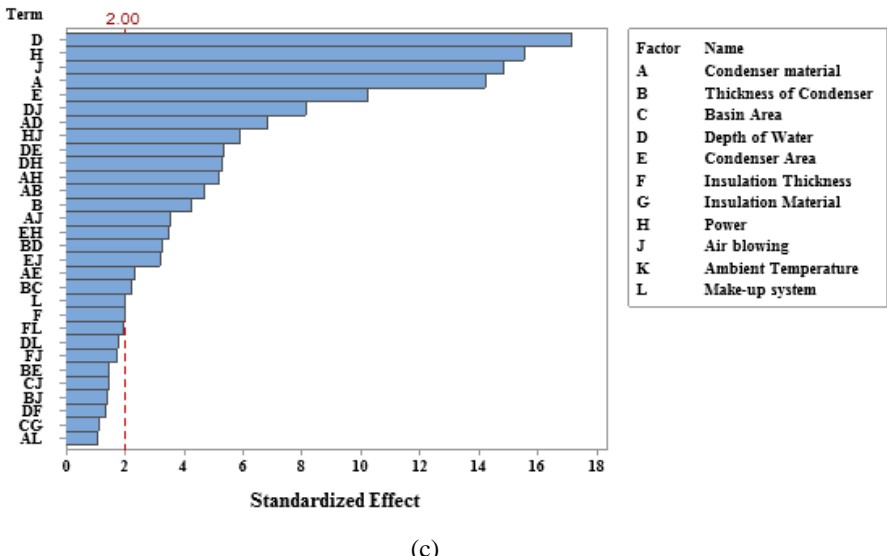


(c)

**Figure 7.** Pareto charts of the standardized effects for **(a)** mass output, **(b)** water
temperature and **(c)** condenser cover temperature.















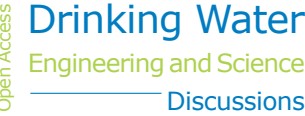

**Table 1:** Description of factor levels.

| Symbol | Factor Name | Low Level | High Level | Unit |
|--------|-------------|-----------|------------|------|
| A | Condenser Material | Glass | Steel | - |
| B | Thickness of Condenser | 4 | 8 | mm |
| C | Basin Area | 0.5 | 1 | $m^2$ |
| D | Depth of Water | 1 | 10 | cm |
| E | Condenser Area | Basin Area | 2*Basin Area | $m^2$ |
| F | Insulation Thickness | 1 | 5 | cm |
| G | Insulation Material | Fiberglass | Wood | - |
| H | Power | 500 | 1000 | Watt |
| J | Air Blowing | Without | With | - |
| K | Ambient Temperature | 20 | 40 | C° |
| L | Make-up Water System | Without | With | - |


**Table 2.** Responses fit values

| Response | Goal | Lower | Target | Upper | Weight | Importance |
|----------|------|-------|--------|-------|--------|------------|
| Tc | Minimum | | 29.238 | 121.323 | 1 | 1 |
| Tw | Maximum | 43.080 | 122.702 | | 1 | 1 |
| Mass | Maximum | 0.306 | 6.474 | | 1 | 1 |




**Table 3.** Values for optimal solar still design

| Solution | Condenser material | Thickness of Condenser | Basin Area | Depth of Water | Condenser Area | Insulation Thickness | Insulation Material |
|----------|--------------------|------------------------|------------|----------------|----------------|----------------------|---------------------|
| 1 | Steel | 0.008 | 1 | 0.01 | Basin area | 0.05 | Fiberglass |




| Solution | Power | Air blowing | Ambient Temperature | Make-up system | Tc Fit | Tw Fit | Mass Fit | Composite Desirability |
|----------|-------|-------------|---------------------|----------------|--------|--------|----------|------------------------|
| 1 | 1000 | With | 40 | Without | 54.2 | 73.5 | 5.9 | 0.635 |
