# Peer review of "Optimization analysis of active solar still using design of experiment method"

_Drinking Water Engineering and Science, 2020_

## Referee Comment (RC1) · Anonymous Referee #1 · 18 Aug 2020

This paper deals with solar still system. The authors used design of experiment (DOE) method to study the effect of many parameters on the performance of active solar still system. Also, to show which parameters have the most significant effect and which of them dose not has any significance. This paper could publish in Drink. Water Eng. Sci. journal, however the following comments must be covered: (1) In the Abstract, the last sentence Line 40 is not clear which parameter is the most influences for the saline water temperature and the condenser cover temperature. This sentence should be rephrased. (2) In the Nomenclatures: # Linees 57 and 58, the symbol (Qcb-w) is repeated two times. # Line 59, to is missed. # Line 62, Ski is English mistake. (3) In the introduction: # Line 79, to is missed in varies from person to another. # Line 96 and 101, The two references Bataineh and Abu Abbas (2020) should be distinguished
by a and b. # Line 105 the dot after Manokar et al (2020) should be removed. The same in case of Khalifa et al (2009) in Line 110. # Line 117, authors did not mentioned anything about the location of the referred system. # Line 125, what is M.S.basin? M.S. should be identified. # Line 136, phase change material should be deleted as you identify PCM in the previous sentence. # The introduction did not cover the papers that deals with study many factors that affect solar still system. Authors should cover this, for instance Poblete et al evaluated the influence of several factors, such as the basin heating, the material of the cover (glass or polycarbonate), the existence of a mirror, the activation of an air extractor, and the existence of a black painted floor in the solar still, in terms of their contribution to brine evaporation. The experiments were conducted with a factorial design approach. "Poblete et al, Investigation of the factors influencing the efficiency of a solar still combined with a solar collector. Desalination and Water Treatment, 57 (2016) 29082-29091." (4) In the Methodology: # The number of lines overlapped with the equations, which did not made these equations unreadable. # Many symbols in the equations did not identified such as Pt. Ab. Tb. mb. CPb, mw, CPw, Qcw-c1. What is the difference between Tb and Tc, and also mb and mw. # In Equation 4, the convection heat transfer from outer condenser cover to sky is written in Nomenclature as Qrc2-s and in Equation 4 as Qrc2-sk, please unify. (5) In Results: # Line 317, Fig. 5, authors used different materials such as glass and steel what is the response in between these two factors represent? How you can consider the material as parameter with a definite value? The same for air blowing, what is the values in between without and with air blowing meant? # In Section 4.1, authors should give explanations for why this factor has the highest effect on the responses. This can be done by comparing with what is found in the literature and to give strong evidences to support their findings. The same is for Section 4.2. Generally, discussion is not sufficient in these two sections. # In Optimization Design Section (Line 384), authors mentioned the conditions in Table 2&3 to achieve the optimal value for the responses. These conditions did not confirmed by any experimental data. This is also did not make any validation for all the results obtained by DOE method. I recommend
to do an experiment with these optimal conditions to compare with the theoretical findings. (6) In the References: # Agrawal et al (2017) is not present in the reference list, while it present in the Introduction section Line 144. # Some references are not given in full such as Manokar et al, the volume and issue no. are missed and Abu Abbas & Al-Abed Allah, the no. of pages is missed. # The reference of Al-harahsheh is repeated two times. # Some references are not given by DOI.

Please also note the supplement to this comment: https://dwes.copernicus.org/preprints/dwes-2020-22/dwes-2020-22-RC1supplement.pdf

---

## Author Comment (AC1) · 19 Aug 2020

Dear Referee, Thanks for considering my article for possible publication in your journal. I would like to thank all the effort made by you and your staff and the referee's thoughtful comments. A revised version was prepared considering all issues that have been raised by the reviewer. kindly find the updated file attachment below.

(1) In the Abstract, the last sentence Line 40 is not clear which parameter is the most influences for the saline water temperature and the condenser cover temperature. This sentence should be rephrased. Author Response: It has been reviewed and modified.

(2) In the Nomenclatures: # Linees 57 and 58, the symbol (Qcb-w) is repeated two times. Author Response It has been reviewed and modified. # Line 59, to is missed.

Author Response It has been reviewed and modified. # Line 62, Ski is English mistake. Author Response It has been reviewed and modified.

(3) In the introduction: # Line 79, to is missed in varies from person to another. Author Response: It has been reviewed and modified.

**Line 96 and 101, The two references Bataineh and Abu Abbas (2020) should be distinguished by a and b. Author Response: It has been reviewed and modified.**

**Line 105 the dot after Manokar et al (2020) should be removed. The same in case of Khalifa et al (2009) in Line 110. Author Response: It has been reviewed and modified.**

**Line 117, authors did not mention anything about the location of the referred system. Author Response: It has been reviewed and modified.**

**Line 125, what is M.S.basin? M.S. should be identified. Author Response: it is a spelling mistake and it has been reviewed and modified.**

**Line 136, phase change material should be deleted as you identify PCM in the previous sentence. Author Response: It has been reviewed and modified.**

**The introduction did not cover the papers that deals with study many factors that affect solar still system. Authors should cover this, for instance Poblete et al evaluated the influence of several factors, such as the basin heating, the material of the cover (glass or polycarbonate), the existence of a mirror, the activation of an air extractor, and the existence of a black painted floor in the solar still, in terms of their contribution to brine evaporation. The experiments were conducted with a factorial design approach. "Poblete et al, Investigation of the factors influencing the efficiency of a solar still combined with a solar collector. Desalination and Water Treatment, 57 (2016) 29082–29091." Author Response: The reference has been added.**

(4) In the Methodology: # The number of lines overlapped with the equations, which did not made these equations unreadable. Author Response: It has been reviewed and modified.

**Many symbols in the equations did not identified such as Pt, Ab, Tb, mb, CPb, mw, CPw, Qcw-c1. What is the difference between Tb and Tc, and also mb and mw. Author Response: The symbole has been identified in the manuscript. Tb is Basin temperature (Co), Tc is Condenser temperature (Co),mb is basin mass (Kg), mw is Inlet water mass (Kg), and mc is Condenser mass (Kg)**

**In Equation 4, the convection heat transfer from outer condenser cover to sky is written in Nomenclature as Qrc2-s and in Equation 4 as Qrc2-sk, please unify. Author Response: It has been reviewed and modified.**

(5) In Results: # Line 317, Fig. 5, authors used different materials such as glass and steel what is the response in between these two factors represent? How you can consider the material as a parameter with a definite value? The same for air blowing, what are the values in between without and with air blowing meant? Author Response: The difference between glass and steel according to their thermal conductivity regarding air blowing The difference between with and without air blowing according to air speed with blowing= 20m/s, without blowing = 0m/s

**In Section 4.1, authors should give explanations for why this factor has the highest effect on the responses. This can be done by comparing with what is found in the literature and to give strong evidences to support their findings. The same is for Section 4.2. Generally, discussion is not sufficient in these two sections. Author Response: The reason behind that can be explained in terms of the evaporation rate. As increasing the amount of external power, the basin water temperature increase. Therefore, the evaporation rate will be increased. Consequently, distilled water is boosted (Ahmed et al 2012). Moreover, as decreasing the basin water depth, the basin water temperature increases faster. Hence, the evaporation rate will be improved, and water productivity is enhanced (Agrawal et al. 2017). Furthermore, when increasing basin water area, the amount of distilled water is increased due to fact that the evaporation rate of the water in the solar still is directly proportional to the exposure area (V. Velmurugan and K. Srithar 2011). Also, as increasing the air speed on the upper condenser layer, the convection**

heat transfer is increased and then the condenser temperature will be decreased (El-Sebaii et al 2004). Author Response for section 4.2 the sufficient explanation has been added

**In Optimization Design Section (Line 384), authors mentioned the conditions in Table 2&3 to achieve the optimal value for the responses. These conditions did not confirmed by any experimental data. This is also did not make any validation for all the results obtained by DOE method. I recommend to do an experiment with these optimal conditions to compare with the theoretical findings. Author Response: we agree with referee that making an experiment with these optimal conditions to compare with the theoretical findings. In future we will conduct the validation to support our results, thank you for your recommendation**

(6) In the References: # Agrawal et al (2017) is not present in the reference list, while it present in the Introduction section Line 144. # Some references are not given in full such as Manokar et al, the volume and issue no. are missed and Abu Abbas & Al-Abed Allah, the no. of pages is missed. # The reference of Al-harahsheh is repeated two times. # Some references are not given by DOI. Author Response: It has been reviewed and modified.

---

## Referee Comment (RC2) · Anonymous Referee #1 · 23 Aug 2020

I found your response but I could not find the revised version of the Manuscript, please submit it.

---

## Author Comment (AC2) · 23 Aug 2020

[revised manuscript text omitted]
. The reason behind that can be explained in terms of the evaporation rate. As increasing the amount of external power, the basin water temperature increase. Therefore, the evaporation rate will be increased. Consequently, distilled water is boosted (Ahmed et al 2012). Moreover, as decreasing the basin water depth, the basin water temperature increases faster. Hence, the evaporation rate will be improved, and water productivity is enhanced (Agrawal et al. 2017). Furthermore, when increasing basin water area, the amount of distilled water is increased due to fact that the evaporation rate of the water in the solar still is directly proportional to the exposure area (V. Velmurugan and K. Srithar 2011). Also, as increasing the air speed on the upper condenser layer, the convection heat transfer is increased and then the condenser temperature will be decreased (El-Sebaii et al 2004). Furthermore, the simulation results indicated that the water depth, the amount of external power, the air blowing system, and the condenser material respectively are the main factors that have the most influence on the water temperature and condenser cover temperature of the system while rest factors have a little effect on it as shown in Fig. 5b. and Fig. 5c.

**4.2 Interaction effect plots:**

The independent variables (factors) might interact with each other. It happens when the influence of one factor depends on the value of another factor. Moreover, the Interaction effects show that a third variable affects the relationship between an independent and dependent factor (responses). This kind of scheme represents the fit values of the dependent factor on the y-axis while the x-axis displays the values of the first independent factor while the different lines describe the values of the second independent factor. About the interaction schemes, parallel lines show that there is no interaction between the two factors while the crossed lines and the lines that will be crossed infer that there is an interaction effect between the factors. Here are the figures for the factors that produced an interaction between each other for various responses. Fig. 6a showed that the interaction effect on mass output. It was clearly noted that (basin area*external power), (basin area*depth of water), (depth of water*external power), (depth of water * air blowing system) and (condenser material *depth of water) respectively have the greatest interaction effect between each other. For example, the scheme for (basin area*external power) explains that mass output level was higher when the external power and the basin area values were high. Conversely, the maximum mass output has been achieved when the external power and the basin area values were low. Fig. 6b showed effect of the interaction on water temperature of the active solar still .it was shown that the highest interaction to produce maximum water temperature were between (depth of water * air blowing system), (condenser material *depth of water), (depth of water*condenser area), (external power * air blowing system) and (depth of water*external power) respectively. For example, the charts for (depth of water*condenser area) and (depth of water*air blowing) describe that the water temperature level is higher at a low level of water depth, and when condenser material and air blowing at the low level also. On the other hand, at a high level of water depth, the water temperature remains as to whether the condenser material and air blowing are at a high or low level. While the interaction plot affected on condenser temperature was described in Fig. 6c. Whereas the important interaction effect were (depth of water * air blowing system), (condenser material *depth of water), (power * air blowing system), (depth of water*condenser area) and (depth of water*external power) respectively.

**4.3 Pareto charts of the standardized effects:**

Fig. 7 display the Pareto charts of the standardized effects for various responses. These charts determine the order of the most significant factors including main and interaction factors that effect on the response's values. It is clearly observed that the most influential factors on mass output are external power, depth of water, and basin area respectively. While in the water temperature and condenser cover temperature, the factors that have most significant effect are depth of water, external power, and air blowing system respectively.

**4.4 Regression equations:**

Regression has been conducted on the results of factorial to show the effects of these factors on the responses values. Eq. (5), Eq. (6), and Eq. (7) are the regression functions predicted from the reduced factorial study which find that the highest and lowest factors affected on three responses: distilled water, saline water temperature and condenser cover temperature respectively. The constant numbers refer to the factors affected ratio while the signals +, - refer to the high or low levels of the factors.

$$
\begin{aligned}
\text{Mass} = {} & -1.026 - 0.0349\,A - 8.1\,B + 0.480\,C + 17.52\,D + 0.0809\,E + 4.67\,F \\
& - 0.0715\,G + 0.000990\,H - 0.1068\,J + 0.00196\,K - 0.1711\,L \\
& + 2.406\,A{*}D - 23.92\,C{*}D + 0.005022\,C{*}H - 0.02169\,D{*}H \\
& + 3.194\,D{*}J + 1.554\,D{*}L
\end{aligned}
\tag{5}
$$

$$
\begin{aligned}
\text{Tw} = {} & 16.72 + 4.36\,A + 3386\,B + 17.19\,C - 10.7\,D - 3.52\,E + 41.5\,F \\
& - 0.627\,G + 0.04329\,H - 4.11\,J + 0.0179\,K - 0.761\,L \\
& - 759\,A{*}B + 1.166\,A{*}E - 0.00571\,A{*}H - 2617\,B{*}C - 13448\,B{*}D \\
& + 58.2\,D{*}E - 0.1492\,D{*}H + 80.9\,D{*}J - 0.00433\,E{*}H + 1.545\,E{*}J - \\
& 0.00675\,H{*}J
\end{aligned}
\tag{6}
$$

$$
\begin{aligned}
\text{Tc} = {} & 10.21 + 3.61\,A + 2095\,B + 0.70\,C + 97.3\,D - 3.20\,E + 50.4\,F \\
& - 0.397\,G + 0.04501\,H - 3.61\,J + 0.0436\,K - 1.013\,L - 1203\,A{*}B \\
& + 77.4\,A{*}D - 0.01053\,A{*}H - 1.815\,A{*}J - 18424\,B{*}D + 60.7\,D{*}E - \\
& 0.2414\,D{*}H + 92.2\,D{*}J - 0.00717\,E{*}H + 1.633\,E{*}J - 0.01207\,H{*}J
\end{aligned}
\tag{7}
$$

**4.5 Optimization Design:**

The designers should create the system by selecting the value of the optimal factors that could enhance mass output. As mentioned above, the maximum water output produced from the solar still could be achieved through increasing the saline water temperature and decreasing the condenser cover temperature. Table. 2 and 3 list the fit values and optimal design selected respectively, to achieve the optimal value for the mass output, saline water temperature and condenser cover temperature.

**5. Conclusion:**

The results of theoretical and statistical analyses of 11 factors on the active solar still system could be summarized as follows:

- The most important factors that can cause increasing in the mass output are the amount of external power, water depth, and the basin area respectively.
- The thickness of the condenser and the ambient air temperature do not affect the mean productivity
- Water depth, the amount of external power, the air blowing system, and the condenser material, respectively, are the main factors that have the most influence on the water temperature of the system.
- (Basin area*power), (basin area*depth of water), (depth of water*power), (depth of water * air blowing system) and (condenser material *depth of water), respectively, have the greatest interaction effect between each other that influence on mass output

- The significant interaction affected on saline water and the condenser temperatures are (depth of water * air blowing system), (condenser material *depth of water), (power * air blowing system), (depth of water*condenser area) and (depth of water*power) respectively.
- The optimal design for the system can be attained is by selecting:
  - Higher external power, basin area, condenser thickness, ambient temperature and insulation thickness.
  - Lower condenser area and depth of water.
  - Using steel condenser material and fiberglass insulations rather than any other materials.
  - Adding air blowing system and removing make-up system.

**Conflict of Interest**

The authors declare that they have no conflict of interest.

**List of figures**

**List of Tables**

[Figure]

**Figure 1.** Solar distillation system.

[Figure]

(a) (b)

**Figure 2. (a)** increasing condensation cover area and **(b)** adding fan to solar still

[Figure]

**Figure 3.** Distilled water cycle system.

[Figure]

**Figure 4.** System flow chart

[Figure]

                                    (a)

[Figure]

                           (b)

[Figure]

                           (C)

**Figure 5.** main effect factors on **(a)** mass output, **(b)** water temperature and **(c)** condenser cover temperature.

[Figure]

                                                (a)

[Figure]

                                                (b)

[Figure]

(c)

**Figure 6.** Interaction effect plot on **(a)** mass output, **(b)** water temperature and **(c)** condenser cover temperature.

[Figure]

(a)

[Figure]

(b)

[Figure]

                                (c)

**Figure 7.** Pareto charts of the standardized effects for **(a)** mass output, **(b)** water
         temperature and **(c)** condenser cover temperature.

**Table 1:** Description of factor levels.

| Symbol | Factor Name | Low Level | High Level | Unit |
|---|---|---|---|---|
| A | Condenser Material | Glass | Steel | - |
| B | Thickness of Condenser | 4 | 8 | mm |
| C | Basin Area | 0.5 | 1 | $m^2$ |
| D | Depth of Water | 1 | 10 | cm |
| E | Condenser Area | Basin Area | 2*Basin Area | $m^2$ |
| F | Insulation Thickness | 1 | 5 | cm |
| G | Insulation Material | Fiberglass | Wood | - |
| H | Power | 500 | 1000 | Watt |
| J | Air Blowing | Without | With | - |
| K | Ambient Temperature | 20 | 40 | C° |
| L | Make-up Water System | Without | With | - |

**Table 2.** Responses fit values

| Response | Goal | Lower | Target | Upper | Weight | Importance |
|---|---|---|---|---|---|---|
| Tc | Minimum | | 29.238 | 121.323 | 1 | 1 |
| Tw | Maximum | 43.080 | 122.702 | | 1 | 1 |
| Mass | Maximum | 0.306 | 6.474 | | 1 | 1 |

**Table 3.** Values for optimal solar still design

| Solution | Condenser material | Thickness of Condenser | Basin Area | Depth of Water | Condenser Area | Insulation Thickness | Insulation Material |
|---|---|---|---|---|---|---|---|
| 1 | Steel | 0.008 | 1 | 0.01 | Basin area | 0.05 | Fiberglass |

| Solution | Power | Air blowing | Ambient Temperature | Make-up system | Tc Fit | Tw Fit | Mass Fit | Composite Desirability |
|---|---|---|---|---|---|---|---|---|
| 1 | 1000 | With | 40 | Without | 54.2 | 73.5 | 5.9 | 0.635 |

---

## Referee Comment (RC3) · Anonymous Referee #1 · 6 Sep 2020

In Optimization Design Section (Line 384), A confirmatory test must be conducted experimentally with the optimal conditions obtained theoretically. This work is not valid without comparing with experimental values. On the other hand, it seems that the DOE method is not based on experimental values, so how the theoretical results are obtained?.

---

## Author Comment (AC3) · 7 Sep 2020

Dear Referee, Thanks for considering my article for possible publication in your journal. I would like to thank all the effort made by you and your staff and the referee's thoughtful comments. A revised version was prepared considering all issues that have been raised by the reviewer.

Reviewer comment

"In Optimization Design Section (Line 384), A confirmatory test must be conducted experimentally with the optimal conditions obtained theoretically. This work is not valid without comparing with experimental values. On the other hand, it seems that the DOE method is not based on experimental values, so how the theoretical results are

obtained?."

Author response

" engineers usually rely on simulation software (such as Matlab, GT series, EES, and etc.) to imitate the operation of any process, and then optimize the design. However, because most of the variables that influence the performance of system are numerical data, there will be infinite variable portfolios. Though the application of simulation software can significantly reduce the experiment cost and enhance the experiment efficiency when each variable has a specific value, it can hardly deal with the situation of infinite potential variable combinations. So, we must consider carefully how to design the experiment plan properly, by which we can identify the key variables and their optimum value intervals with a few experiments as possible. To solve this problem, in this paper, we introduce the DOE (Design of Experiment) methodology to arrange and conduct the experiments rationally in the future with optimal conditions. With the statistical analysis of simulational experiment data, we set up the functional relationship model, discuss about the influences of different variables on the performance of the system, then search out and test the combination of the optimum variable in the future."

We also explain previous points in the Methodology specifically at section [2.3 Design of Experimental] inline 237 in our manuscript.

Furthermore, we have 11 variables and 2 levels for each one, so we need $2^{11}$ setups and this will take a lot of time and cost. Therefore, it is impossible to apply experimnt setup in our study. So the purpose of our study is to determine the significant and insignificant factors to achieve the experiment in the future with optimal conditions.

---

## Referee Comment (RC4) · Anonymous Referee #2 · 11 Nov 2020

I believe that the present paper contains interesting results. Therefore, I think the manuscript could be accepted for publication after the following major issues are addressed. The whole manuscript revised once again - it has many grammatical, stylistic and typographical mistakes. English should be checked by native speaker. Concrete comments are shown as follows. Abstract Abstracts of the paper lacks important information. Abstract provides detailed explanation of the methods weather results and conclusions of an abstract are lacking.

Introduction 1) English should be checked by the native speaker 2) Solar still should be introduce in the second paragraph of the introduction. 3) References to solar still performance should be added. What is the performance of the solar still and why should be improved? 4) What happens with the solar still if you increase condenser cover

temperature and increase saline water temperature? 5) Line 90-91 - "Enhancing the productivity of solar still has received significant attention from many researchers." – What researchers? References should be added 6) Same for line 92-95 7) Line 105 typo "Manokar et al (2020). Analyzed…." 8) Line 155-157 - "It should be remarked that all of the researches have studied the influence of utilizing one parameter at a time while keeping the other parameters fixed will not occur to understand the interaction." This sentence is really not understandable. What is will not occur to understand the interaction? 9) Line 158 "we collected all the parameters that could affect the active solar still" – My suggestion is to name all main parameters that you took into consideration. If not in introduction then certainly in 2.3 Design of Experimental 10) Line 320-322 – "The results found that the most important factors that enhance mass output are amount of external power, water depth, and basin area respectively." It would be good to support that information with findings from the literature and explanation why is that so.

---

## Author Comment (AC4) · 14 Nov 2020

Dear Editor and Reviewer, The authors will like to appreciate the reviewers for taking the time to constructively critique the manuscript in order to enhance its quality suitable for the research community.

Comment 1: English should be checked by the native speaker Author Response: The article has gone through detailed and thorough proofreading by a native English-speaking person and has also been checked for any punctuation as well as grammatical error using an English correction checker (Grammarly). Action: It has been reviewed and modified.

Comment 2: Solar still should be introduced in the second paragraph of the introduction Author Response: The authors welcome the suggestions from the reviewer. The updated manuscript has been prepared to reflect the suggestions by the reviewer accordingly. Action: It has been reviewed and modified

Comment 3: References to solar still performance should be added. What is the performance of the solar still and why should be improved? Action: It has been reviewed and modified

Comment 4: What happens with the solar still if you increase condenser cover temperature and increase saline water temperature? Action: Increasing different temperature between the salty water and condenser cover enhance the evaporation rate and condensation process which lead to increasing productivity (different temperature is driving force)

Comment 5: Line 90-91 - "Enhancing the productivity of solar still has received significant attention from many researchers." – What researchers? References should be added Action: It has been reviewed and added

Comment 6: Same for line 92-95 Action: It has been reviewed and added

Comment 7: Line 105 typo "Manokar et al (2020). Analyzed. . .." Action: The correction has been made and highlighted in yellow, as suggested by the reviewer in the updated manuscript.

Comment 8: Line 155-157 - "It should be remarked that all of the researchers have studied the influence of utilizing one parameter at a time while keeping the other parameters fixed will not occur to understand the interaction." This sentence is really not understandable. What is will not occur to understand the interaction? Author Response: The authors appreciate the reviewer for his critical comment on the manuscript. The updated manuscript has been prepared to reflect the suggestion by the reviewer accordingly Action: It has been reviewed and modified

Comment 9: Line 158 "we collected all the parameters that could affect the active solar still" – My suggestion is to name all main parameters that you took into consideration. If not in the introduction then certainly in 2.3 Design of Experimental. Author Response: The authors are grateful to the reviewer for this observation, and in fact, we agree with the suggestion Action: It has been reviewed and added

Comment 10: ) Line 320-322 – "The results found that the most important factors that enhance mass output are amount of external power, water depth, and basin area respectively." It would be good to support that information with findings from the literature and an explanation of why is that so.

Author Response: The authors are grateful to the reviewer for such an in-depth review. The updated manuscript captures the suggestion by the reviewer Action: It has been reviewed and added

**Supplement:**

**Optimization analysis of active solar still using design of experiment method**

Mohammad Omar Abu Abbas[a], Malik Yousef Al-Abed Allah[b*,] Qais Nedal Al-Oweiti[a]

   [a] Department of Mechanical Engineering, Jordan University of Science and
Technology, Irbid, Jordan

   [b] Department of Mechanical Engineering, Al-Huson University College, Al-Balq'a
Applied University, Irbid, Jordan

*Correspondence: *malek.abedallah@bau.edu.jo*

**Mohammad Omar Abu Abbas,** Department of Mechanical Engineering, Jordan
University of Science and Technology, 22110, Irbid, Jordan,
*moabuabbas16@eng.just.edu.jo*

**Malik Yousef Al-Abed Allah,** Department of Mechanical Engineering, Al-Huson
University College, Al-Balq'a Applied University, Jordan, 21510, Irbid, Jordan,
*malek.abedallah@bau.edu.jo*

**Qias Nedal Al-Oweiti**, Department of Mechanical Engineering, Jordan University of
Science and Technology, 22110, Irbid, Jordan, *qnaloweiti16@eng.just.edu.jo*

**Key words:** solar still, DOE, factorial design, thickness, productivity, water depth, insulation.

**Abstract:**

The mathematical model for different configurations of active solar still has been analyzed. Theoretical analysis of energy balance for the active solar still components has been developed. A statistical manner for examination, evaluation, and optimizing the performance of the active solar distillation system with known input factors has been performed using the Design of Experiments (DOE) method. Some processes with input variables (factors) and predicted output variables (responses) have been evaluated. Input factors influencing the responses have been identified. The impact of each variable (factor) and the integration of two factors at the same time (called interactions) have been estimated. Influences of various factors on a particular study at a time rather than performing different separated studies have been investigated. 11 variables (basin area, depth of saline water, external power, air blowing system, condenser material, condenser thickness, condenser area, insulation thickness, insulation material, ambient air temperature, and make-up water system ) have been studied to show their effects on three responses (mass output, saline water temperature and condenser cover temperature). The statistical results showed that the most significant factors affected mass output (distilled water) were the external power, the depth of the saline water, and the basin area of the active still, respectively. Furthermore, the most influential factors affecting the saline water temperature and the condenser cover temperature were the depth of saline water, external power, and air blowing system respectively.

**Nomenclatures**:

$A_p$       basin Area ($m^2$)

$cp_b$     Basin Specific heat (J/kg.k)

$cp_w$     Water Specific heat (J/kg.k)

$cp_c$     Condenser Specific heat (J/kg.k)

$P_t$       External power (W/m$^2$)

$Q_{cb-w}$     Convection heat transfer from basin plate to saline water (W)

$Q_{cw-c1}$     Convection heat transfer from saline water to the condenser (W)

$Q_{rw-c1}$     Radiation heat transfer from saline water to the inner condenser (W)

$Q_{ew-c1}$     Evaporation heat transfer from saline water to the inner condenser (W)

$Q_{cw-c1}$     Convection heat transfer from saline water to the inner condenser (W)

$Q_{cc2-a}$     Convection heat transfer from outer condenser cover to ambient (W)

$Q_{rc2-sk}$     Convection heat transfer from outer condenser cover to the sky (W)

$Q_{cnc1-c2}$     Conduction heat transfer from inner condenser cover to the outer condenser (W)

$Q_{loss-ba}$     Conduction heat transfer from basin plate to ambient (W)

$Q_{mw}$     Make-up saline water (W)

$T_b$       Basin temperature (C$^o$)

$T_c$       Condenser temperature (C$^o$)

$T_w$       Water temperature (C$^o$)

$m_b$       basin mass (Kg)

$m_w$       Inlet water mass (Kg)

$m_c$       Condenser mass (Kg)

**1. Introduction:**

Water is an essential component of human health. Nearly 60% of the human body is composed of water. It is important to note that the individual's need for water varies from person to another depending on the nature of the individual's daily physical activities and the drought proportion in the place where they live. Therefore, individuals tend to drink sufficient amounts of water to prevent them from the drought. Consequently, it leads to drain the body's energy, and cause tired. The National Academy of Sciences has determined the amount of water that is recommended daily, namely 3.7 liters of water for males and 2.7 liters of water for females. In fact, these amounts include water obtained from drinking water, and eating other foods and beverages. Although three-quarters of the earth is covered with water but, the clean water does not exceed 2.75%, which is a low proportion comparing with saltwater.

solar still is a green energy product that utilizes the natural energy of the sun to purify water. The solar-still process uses the sun instead of other sources such as fossil fuels to gain the energy needed for purification. Solar stills are then able to provide distilled water for cooking and drinking, even in areas where there are no other sources of energy, while still being friendly to the environment. The solar stills are broadly classified into two types namely, passive and active solar stills. active solar still uses some external setup like external power to feed an extra thermal energy for faster evaporation While Passive solar stills evaporate the basin water directly through sun. Design modifications of Active solar stills include solar still integrated with solar concentrators, solar still integrated with solar heater, and solar still with heat exchanger While passive solar stills include spherical solar still, wick type stills, etc.

The daily production and efficiency of a conventional solar still are relatively low. In best optimized operating conditions, daily production and efficiency are about 3-5 $L/m^2$.day and 30-45% respectively.[A. E. Kabeel and S. A. El-Agouz (2011)]. Numbers of attempts have been made to look for ways to inhace its efficiency and productivity. For more detail of the effective parameters of a conventional solar still, readers are referred to [Abujazar et al (2016), Sharshir et al (2016), Chandrashekara et al (2017)].

Improving the performance of solar still depends mainly on decreasing condenser cover temperature and increasing saline water temperature. Enhancing the productivity of solar still has received significant attention from many researchers. ( Bataineh and Abu Abbas[a]; Manokar et al (2020) , Khalifa et al (2009); Zurigat et al. (2004), Madhlopa et al. (2009), The daily production of solar still depends on several factors such as climatic conditions [solar radiation intensity, ambient temperature, and wind speed], (Omar et al 2017, el-sebaii 2000) condensation surface inclination, (Bilal

et al 2000) insulation type and thickness, (Manokar et al 2020)  solar still geometry, ( Mehrzad et al 2017) the orientation of still and depth of salty water, ( Al-abed Allah and Abu Abbas 2020)

[revised manuscript text omitted]

**4. Results:**

The results of mathematical and designing calculations could discover the effect of different factors on active solar still responses. Three responses have been studied: the amount of distilled water (mass output), water temperature, and condenser cover temperature. External power, basin area, water depth, insulation material, insulation thickness, condenser material (according to material thermal conductivity value), condenser area, thickness of condenser, air blowing system according to air speed (without air blowing = 0 m/s and with air blowing = 20 m/s) , Make-up water system, and ambient temperature are considered as variables to understand their influences on the mentioned responses. To be more effective, the simulation results were gained based on the design of the experiment approach (DOE). The (DOE) was conducted using a reduced factorial method to show their direct effects, their interactions, and the optimization design for the system.

**4.1 Main effect plots on the responses:**

Fig. 5a, Fig. 5b and Fig. 5c showed the main factors influenced on the responses of active solar still system. The x-axis shows responses values while the y-axis shows the high and the low levels of the factors. It was clearly noted that, as inclination of the lines increase, the effect of the factors on the responses will be significant. The results found that the most important factors that enhance mass output are the amount of external power, water depth, and basin area respectively. Where the mean mass output recorded at the high and low levels were 3.02 L and 1.24 L respectively for external power factor and 1.3L and 2.8L respectively for water depth factor. While, it is reached about 2.8L and 1.4L at high and low levels of the basin area respectively. Moreover, other factors have little effect on the system. The reason behind that can be explained in terms of the evaporation rate. As increasing the amount of external power, the basin water temperature increase. Therefore, the evaporation rate will be increased. Consequently, distilled water is boosted (Ahmed et al 2012). Moreover, as

decreasing the basin water depth, the basin water temperature increases faster. Hence, the evaporation rate will be improved, and water productivity is enhanced (Agrawal et al. 2017). Furthermore, when increasing basin water area, the amount of distilled water is increased due to fact that the evaporation rate of the water in the solar still is directly proportional to the exposure area (V. Velmurugan and K. Srithar 2011). Also, as increasing the air speed on the upper condenser layer, the convection heat transfer is increased and then the condenser temperature will be decreased (El-Sebaii et al 2004). Furthermore, the simulation results indicated that the water depth, the amount of external power, the air blowing system, and the condenser material respectively are the main factors that have the most influence on the water temperature and condenser cover temperature of the system while the rest factors have a little effect on it as shown in Fig. 5b. and Fig. 5c.

**4.2 Interaction effect plots:**

The independent variables (factors) might interact with each other. It happens when the influence of one factor depends on the value of another factor. Moreover, the Interaction effects show that a third variable affects the relationship between an independent and dependent factor (responses). This kind of scheme represents the fit values of the dependent factor on the y-axis while the x-axis displays the values of the first independent factor while the different lines describe the values of the second independent factor. About the interaction schemes, parallel lines show that there is no interaction between the two factors while the crossed lines and the lines that will be crossed infer that there is an interaction effect between the factors. Here are the figures for the factors that produced an interaction between each other for various responses. Fig. 6a showed that the interaction effect on mass output. It was clearly noted that (basin area*external power), (basin area*depth of water), (depth of water*external power), (depth of water * air blowing system) and (condenser material *depth of water) respectively have the greatest interaction effect between each other. For example, the scheme for (basin area*external power) explains that the mass output level was higher when the external power and the basin area values were high. Conversely, the maximum mass output has been achieved when the external power and the basin area values were low. Fig. 6b showed the effect of the interaction on the water temperature of the active solar still .it was shown that the highest interaction to produce maximum water temperature was between (depth of water * air blowing system), (condenser material *depth of water), (depth of water*condenser area), (external power * air blowing system) and (depth of water*external power) respectively. For example, the charts for (depth of water*condenser area) and (depth of water*air blowing) describe that the water temperature level is higher at a low level of water depth, and when condenser material and air blowing at the low level also. On

the other hand, at a high level of water depth, the water temperature remains as to whether the condenser material and air blowing are at a high or low level. While the interaction plot affected on condenser temperature was described in Fig. 6c. Whereas the important interaction effect was (depth of water * air blowing system), (condenser material *depth of water), (power * air blowing system), (depth of water*condenser area) and (depth of water*external power) respectively.

**4.3 Pareto charts of the standardized effects:**

Fig. 7 display the Pareto charts of the standardized effects for various responses. These charts determine the order of the most significant factors including main and interaction factors that effect on the response's values. It is clearly observed that the most influential factors on mass output are external power, depth of water, and basin area respectively. While in the water temperature and condenser cover temperature, the factors that have the most significant effect are depth of water, external power, and air blowing system respectively.

**4.4 Regression equations:**

Regression has been conducted on the results of factorial to show the effects of these factors on the response values. Eq. (5), Eq. (6), and Eq. (7) are the regression functions predicted from the reduced factorial study which found that the highest and lowest factors affected on three responses: distilled water, saline water temperature and condenser cover temperature respectively. The constant numbers refer to the factors affected ratio while the signals +, - refer to the high or low levels of the factors.

$$
\begin{aligned}
\text{Mass} = {} & -1.026 - 0.0349\,A - 8.1\,B + 0.480\,C + 17.52\,D + 0.0809\,E + 4.67\,F \\
& - 0.0715\,G + 0.000990\,H - 0.1068\,J + 0.00196\,K - 0.1711\,L \\
& + 2.406\,A*D - 23.92\,C*D + 0.005022\,C*H - 0.02169\,D*H \\
& + 3.194\,D*J + 1.554\,D*L
\end{aligned}
$$

$$(5)$$

$$
\begin{aligned}
\text{Tw} = {} & 16.72 + 4.36\,A + 3386\,B + 17.19\,C - 10.7\,D - 3.52\,E + 41.5\,F \\
& - 0.627\,G + 0.04329\,H - 4.11\,J + 0.0179\,K - 0.761\,L \\
& - 759\,A*B + 1.166\,A*E - 0.00571\,A*H - 2617\,B*C - 13448\,B*D \\
& + 58.2\,D*E - 0.1492\,D*H + 80.9\,D*J - 0.00433\,E*H + 1.545\,E*J - \\
& 0.00675\,H*J
\end{aligned}
$$

$$(6)$$

$$
\begin{aligned}
\text{Tc} = {} & 10.21 + 3.61\,A + 2095\,B + 0.70\,C + 97.3\,D - 3.20\,E + 50.4\,F \\
& - 0.397\,G + 0.04501\,H - 3.61\,J + 0.0436\,K - 1.013\,L - 1203\,A*B \\
& + 77.4\,A*D - 0.01053\,A*H - 1.815\,A*J - 18424\,B*D + 60.7\,D*E - \\
& 0.2414\,D*H + 92.2\,D*J - 0.00717\,E*H + 1.633\,E*J - 0.01207\,H*J
\end{aligned}
$$

$$(7)$$

**4.5 Optimization Design:**

The designers should create the system by selecting the value of the optimal factors that could enhance mass output. As mentioned above, the maximum water output produced from the solar still could be achieved by increasing the saline water temperature and decreasing the condenser cover temperature. Table. 2 and 3 list the fit values and optimal design selected respectively, to achieve the optimal value for the mass output, saline water temperature and condenser cover temperature.

**5. Conclusion:**

The results of theoretical and statistical analyses of 11 factors on the active solar still system could be summarized as follows:

- The most important factors that can cause increase in the mass output are the amount of external power, water depth, and the basin area respectively.
- The thickness of the condenser and the ambient air temperature do not affect the mean productivity
- Water depth, the amount of external power, the air blowing system, and the condenser material, respectively, are the main factors that have the most influence on the water temperature of the system.
- (Basin area*power), (basin area*depth of water), (depth of water*power), (depth of water * air blowing system) and (condenser material *depth of water), respectively, have the greatest interaction effect between each  other that influence the mass output
- The significant interaction affected on saline water and the condenser temperatures are (depth of water * air blowing system), (condenser material *depth of water), (power * air blowing system), (depth of water*condenser area) and (depth of water*power) respectively.
- The optimal design for the system can be attained is by selecting:
    - Higher external power, basin area, condenser thickness, ambient temperature and insulation thickness.
    - Lower condenser area and depth of water.
    - Using steel condenser material and fiberglass insulations rather than any other materials.
    - Adding an air blowing system and removing the make-up system.

**Conflict of Interest**

The authors declare that they have no conflict of interest.

613

[Figure]

614

615  **Figure 4.** System flow chart

616

617

618

619

620

621                                             (a)

622

623                                              (b)

624

625

626                                              (C)

**Figure 5.** main effect factors on **(a)** mass output, **(b)** water temperature and **(c)** condenser cover temperature.

[Figure]

629                                              (a)

[Figure]

630                                              (b)

631

632

633
634                 (c)

635 **Figure 6.** Interaction effect plot on **(a)** mass output, **(b)** water temperature and **(c)**
636                 condenser cover temperature.

637

638

639

640

641

642

643

644

645

646

647
648                                                    (a)

649

650

651
652                                                    (b)

[Figure]

653
654                                                    (c)

Figure 7. Pareto charts of the standardized effects for (a) mass output, (b) water
temperature and (c) condenser cover temperature.

657

658

659

660

661

662

663

664

665

666

667

668

669

**Table 1:** Description of factor levels.

| Symbol | Factor Name | Low Level | High Level | Unit |
|---|---|---|---|---|
| A | Condenser Material | Glass | Steel | - |
| B | Thickness of Condenser | 4 | 8 | mm |
| C | Basin Area | 0.5 | 1 | $m^2$ |
| D | Depth of Water | 1 | 10 | cm |
| E | Condenser Area | Basin Area | 2*Basin Area | $m^2$ |
| F | Insulation Thickness | 1 | 5 | cm |
| G | Insulation Material | Fiberglass | Wood | - |
| H | Power | 500 | 1000 | Watt |
| J | Air Blowing | Without | With | - |
| K | Ambient Temperature | 20 | 40 | C° |
| L | Make-up Water System | Without | With | - |

**Table 2.** Responses fit values

| Response | Goal | Lower | Target | Upper | Weight | Importance |
|---|---|---|---|---|---|---|
| Tc | Minimum | | 29.238 | 121.323 | 1 | 1 |
| Tw | Maximum | 43.080 | 122.702 | | 1 | 1 |
| Mass | Maximum | 0.306 | 6.474 | | 1 | 1 |

**Table 3.** Values for optimal solar still design

| Solution | Condenser material | Thickness of Condenser | Basin Area | Depth of Water | Condenser Area | Insulation Thickness | Insulation Material |
|---|---|---|---|---|---|---|---|
| 1 | Steel | 0.008 | 1 | 0.01 | Basin area | 0.05 | Fiberglass |

| Solution | Power | Air blowing | Ambient Temperature | Make-up system | Tc Fit | Tw Fit | Mass Fit | Composite Desirability |
|---|---|---|---|---|---|---|---|---|
| 1 | 1000 | With | 40 | Without | 54.2 | 73.5 | 5.9 | 0.635 |